# An integrated photonic engine for programmable atomic control

Ian Christen [1] ✉, Thomas Propson[1], Madison Sutula [1], Hamed Sattari[2], Gregory Choong[2], Christopher Panuski [1], Alexander Melville[3], Justin Mallek[3], Cole Brabec[1], Scott Hamilton [3], P. Benjamin Dixon [3], Adrian J. Menssen[1], Danielle Braje [3], Amir H. Ghadimi[2] & Dirk Englund [1] ✉

Solutions for scalable, high-performance optical control are important for the development of scaled atom-based quantum technologies. Modulation of many individual optical beams is central to applying arbitrary gate and control sequences on arrays of atoms or atom-like systems. At telecom wavelengths, miniaturization of optical components via photonic integration has pushed the scale and performance of classical and quantum optics far beyond the limitations of bulk devices. However, material platforms for high-speed telecom integrated photonics lack transparency at the short wavelengths required by leading atomic systems. Here, we propose and implement a scalable and reconfigurable photonic control architecture using integrated, visible-light modulators based on thin-film lithium niobate. We combine this system with techniques in free-space optics and holography to demonstrate multi-channel, gigahertz-rate visible beamshaping. When applied to silicon-vacancy artificial atoms, our system enables the spatial and spectral addressing of a dynamically-selectable set of these stochastically-positioned point emitters.

Controlling coherent light is essential to the use and understanding of atomic systems[1–4]. A number of features are desired in optical control: the application of frequency shifting[5] or frequency-domain shaping[6,7]; the execution of fast operations, for instance to compensate for finite atom lifetime[8]; and the precise delivery of programmable optical phase and amplitude profiles. Together, these features comprise the ideal of spectro-temporal control over an optical mode, where light is manipulated across frequency and time with precision. The switching bandwidth of an optical modulator defines the extent and speed at which a spectro-temporal waveform can be realized.

Large-scale programmable quantum information processing on atomic systems requires the implementation of spectro-temporal control on individual spatially-distributed optical modes corresponding to atomic sites[9] (Fig. 1). Previous work demonstrating multi-channel atomic addressing has largely involved extending bulk acousto-optic (AO) technologies—limited to $\mathcal{O}(\text{GHz})$ switching bandwidth—to multiple spatial channels, whether by mapping frequency domain signals to spatial sites via AO deflectors[10–13] or by arraying many AO modulators[14,15]. Site count and modulation speed for AO deflectors are currently limited by acoustic velocity and bandwidth, especially when used in a rasterized mode, and the frequency gradient present in AO-deflected patterns is problematic for frequency-sensitive protocols without additional complexity[13]. Arrayed bulk modulators face the problem of scaling beyond the tens of channels demonstrated and towards the thousands or more necessary to realize fault-tolerant quantum computation[16]. The complexity of assembling such devices scales with channel count, making growth by these orders of magnitude impractical. A similar challenge led to the development of integrated classical computing technologies which transcended the limitations of bulk electronics via miniaturization of components and parallelization of fabrication[17].

Likewise, integrated photonics enables scales and capabilities surpassing bulk optics[18–24]. However, photonic integration at visible wavelengths has largely been passive, in part because silicon nitride

[1]Research Laboratory of Electronics, Massachusetts Institute of Technology, Cambridge, MA, USA. [2]Centre Suisse d'Electronique et de Microtechnique (CSEM), Neuchâtel, Switzerland. [3]Lincoln Laboratory, Massachusetts Institute of Technology, Lexington, MA, USA. ✉e-mail: ichr@mit.edu; englund@mit.edu

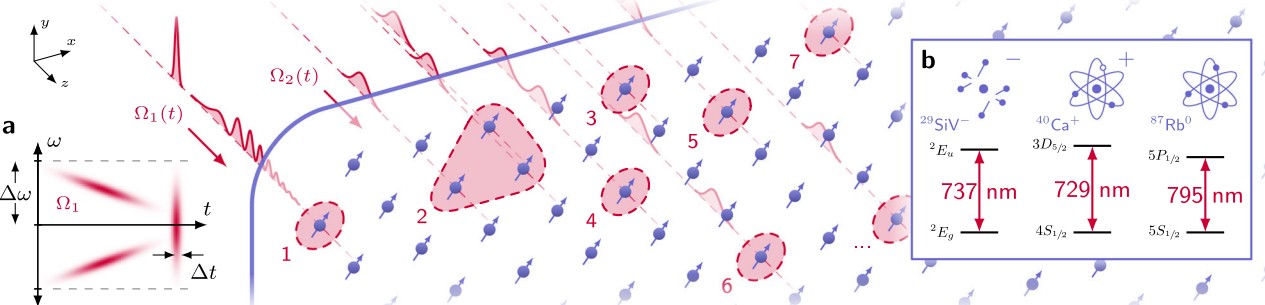

**Fig. 1 | Multi-channel optical addressing of atomic systems.** Individually-controlled optical beams are incident upon single or grouped atomic systems, providing programmable control. **a** An example spectrogram of a waveform $\Omega_1$ consisting of a chirped pulse followed by a fast pulse incident on the channel labeled "1". Large switching bandwidth $\Delta\omega$ and, correspondingly, small switching time $\Delta t$ are desired for flexible spectro-temporal control. **b** Numerous atomic systems—from solid state memories, to ions, to neutral atoms—have visible wavelength transitions near 780 nm, a regime inaccessible by active photonics developed in silicon or indium phosphide for telecommunications. We note a canonical example for each category, including especially the negatively-charged silicon-vacancy center in diamond which we use in "Application" to demonstrate the capabilities of our device.

(SiN), the prevailing material for visible photonics, does not have a strong intrinsic electro-optic effect[25]. While methods to add fast active modulation to SiN or similar passive materials have been investigated, none combine large and broadband switching bandwidth (DC to few GHz) with small switching voltages (<5 V)[26–32]. Low switching voltages are important for direct compatibility with scalable high-speed complementary metal-oxide semiconductor (CMOS) electronics operating at $\mathcal{O}(1\text{ V})$[33].

Thin-film lithium niobate (TFLN) has recently risen as an excellent platform for integrated nonlinear optics[34–36]. TFLN combines large switching bandwidth and small switching voltage, with the state-of-the-art pushing above 100 GHz[34] and below one volt[37,38]. Moreover, lithium niobate possesses a wide bandgap with transparency down to 350 nm, allowing the visible-wavelength operation[38–40] which is critical for addressing many atomic systems. Significant work over the past decade has made TFLN technologically ready to reliably fabricate large-scale circuits at wafer scales[41,42]; however, to date, no large-scale multi-mode systems have been developed due to prevailing challenges in uniformity and stabilization as part of a system.

Here, we introduce an architecture to overcome these limitations, enabling scalable multi-mode and multi-GHz-rate integrated visible light control with excellent cross-channel uniformity, precision stabilization, and reconfigurable projective mapping between modulated channels and atomic targets.

## Results
### Architecture
Our photonic engine consists of sixteen TFLN Mach-Zehnder interferometers (MZIs), routed to input ($1\times16$) and output ($4\times4$) grating banks at one side of the chip (Fig. 2c, d). Each grating couples a waveguide mode at 780 nm into a free-space vertical Gaussian beam. We measure channel insertion losses of ~20 dB at 780 nm, which we attribute to grating inefficiencies associated with our conservative choice of design parameters while co-developing a wafer-scale TFLN foundry process. While edge-couplers with $\mathcal{O}(\text{dB})$ loss to free-space were also fabricated for testing, our experiments focused on this 2D grating array to emulate the scalable geometry of megapixel-aperture cameras, CCDs, and SLMs ("Chip layout"). Each MZI amplitude modulator consists of two directional couplers and 3-mm-long phase shifters in push-pull ground-signal-ground configuration. The MZIs share grounds and are wirebonded to a printed circuit board (PCB; Fig. 2f) for electrical control. At 780 nm, we measure CMOS-compatible switching voltages of $V_\pi \sim 2.2$ V, extinction ratios exceeding 20 dB, and a PCB-limited 3 dB switching bandwidth of 7 GHz (Fig. 2g, h).

We use a low-bandwidth commercial liquid crystal on silicon (LCoS) spatial light modulator (SLM) to uniformly fanout optical power to our high-bandwidth integrated photonic channels. Compared to the serial losses and associated scaling limits characteristic of on-chip routing matrices, this technique enables near-lossless free-space reconfigurability of thousands of input modes. The SLM produces a static hologram of beamspots coupled to the input grating couplers through the reflection port of a polarizing beamsplitter (Fig. 2a, c). The orthogonal rotation of the output gratings relative to the input gratings (Fig. 2d) couples modulated light through the other port of the beamsplitter and towards the target atomic systems. We monitor the output power of each channel with a camera and apply weighted Gerchberg-Saxton (WGS) feedback[43] on the fanout hologram to refine cross-channel output uniformity, correcting for hologram alignment errors and variations in channel insertion loss. A dozen iterations of WGS feedback yield uniformities better than 1% (Fig. 2d ii, e). To our knowledge, this is the first application of WGS to optical fanout for integrated photonics. The $\mathcal{O}(10^6)$ stable degrees of freedom in commercial SLMs permit a greater level of fanout precision and reliability while simultaneously circumventing the losses of on-chip-photonic or fiber-based splitters which scale exponentially with tree depth.

### Stabilization
Lithium niobate is known to be susceptible to zero-point drift, where the voltage corresponding to the point of highest extinction varies over time, which is attributed to fluctuations in charges trapped in the phase shifters[44–47]. In our experiment, for example, we observe zero-point drift on the time scale of several seconds when monitoring a free-running, open-loop configuration. Photodiode feedback and analog stabilization circuitry typically mitigate this drift for bulk modulators, at the expense of scalability. To scale beyond hundreds of channels, we therefore developed a closed-loop and camera-based parallel feedback scheme to stabilize each channel's electrical degree of freedom. Monitoring the optical channels with the same camera used for WGS optimization, we measure the transmission of each channel at four voltages: two pairs, each equally spaced around estimates for the modulator's minimum (LO) or maximum (HI) setpoint (Fig. 3b–c). Closed-loop locking our setpoints to voltages with balanced pairwise transmission yields alignment with the locally-quadratic points of minimum or maximum transmission (Fig. 3c), while avoiding the challenge of directly measuring attenuated signals (i.e. LO) with short integration times. We operate this locking loop at about 200 Hz, much faster than the $\mathcal{O}(\text{s})$ timescale of drift (see "Closed-loop zero-point stabilization" for more details).

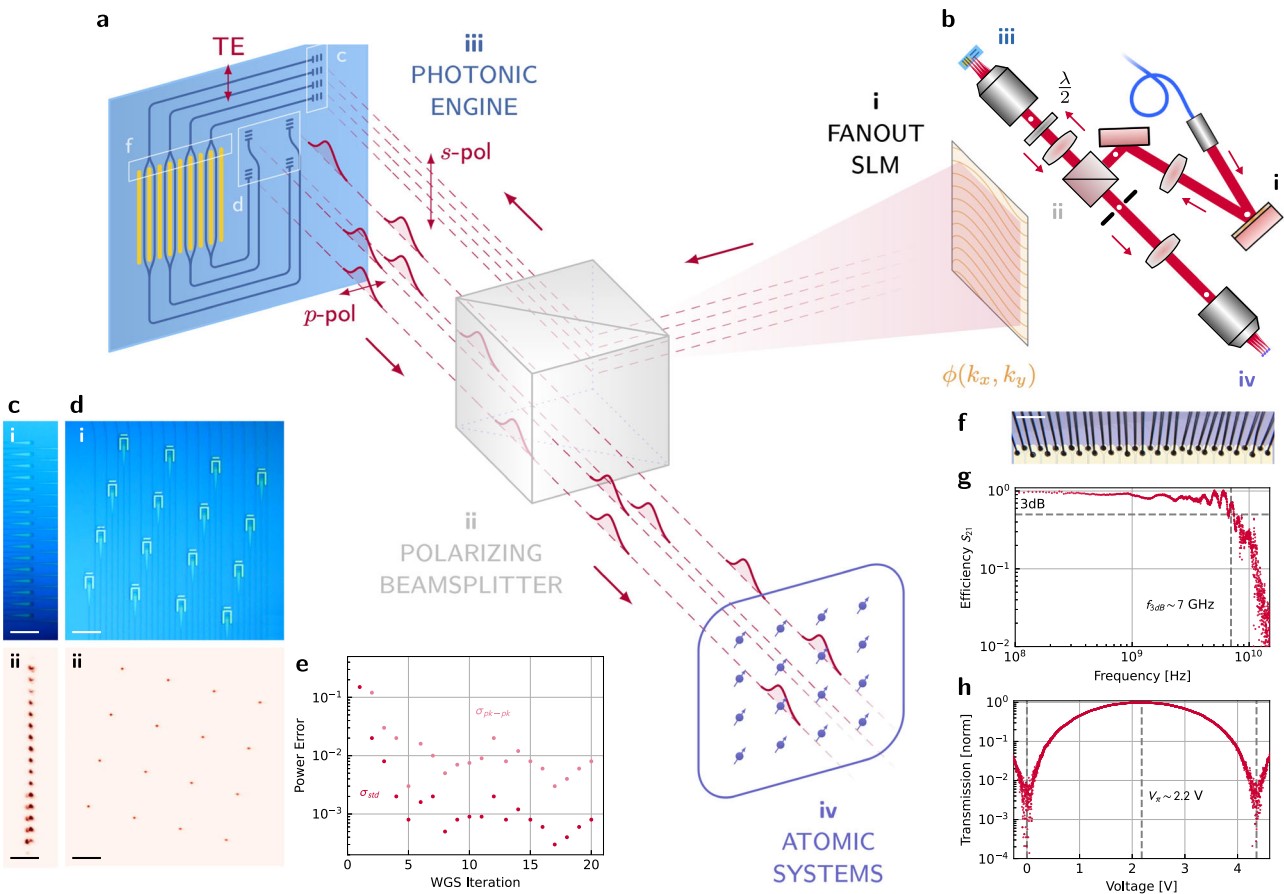

**Fig. 2 | Architecture, optical fanout, and modulator performance. a** An incident *s*-polarization hologram produced by a Fourier-domain SLM couples to many transverse electric (TE) waveguide modes via grating couplers (**i**–**iii**; **c**). Integrated MZIs impart programmed waveforms on each channel. Output grating couplers direct light, now *p*-polarized, back to free-space towards the target sites (**iv**). For better visibility, four channels–rather than the full sixteen–are shown in this diagram. **b**, A top-down view of our setup illustrates the lenses and objectives omitted from the simplified diagram in **a**. The half-waveplate in front of the objective aligns the beamsplitter and grating polarization axes. A double-4*f* lens configuration, with one lens participating in both 4*fs*, enables large fields of view. White dots represent imaging or Fourier planes. **c** The 1 × 16 input and **d**, the 4 × 4 output grating coupler arrays on our device in white light (**i**) and under coherent excitation (**ii**), showing the WGS-generated fanout hologram (**c ii**) and the resulting uniform output beams (**d ii**). Scalebars represent 50 μm. **e** Several iterations of WGS yield peak-to-peak optical output power errors $\sigma_{\text{pk-pk}}$ of ~1% and standard deviations $\sigma_{\text{std}}$ of ~0.1%. **f** Wirebonding to a PCB connects each of our sixteen channels to external control electronics. The scalebar represents 500 μm. **g**, Switching bandwidth of a modulator channel. **h** A trace of device transmission versus voltage.

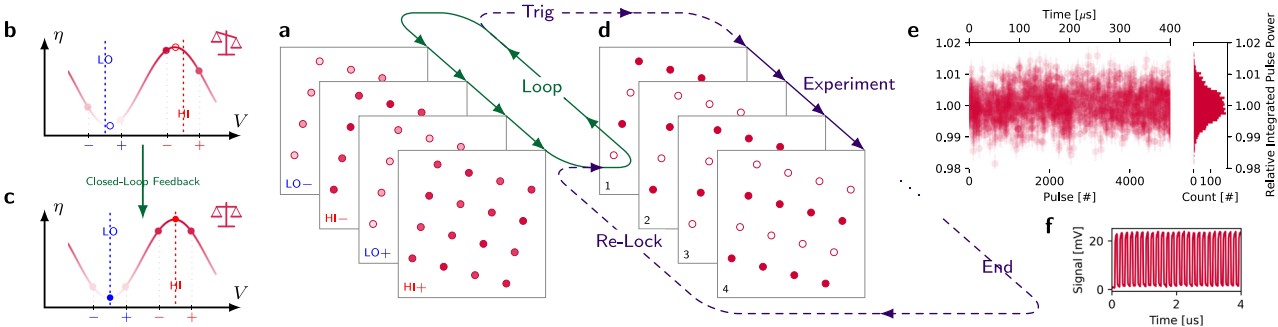

**Fig. 3 | Stabilizing zero-point drift. a** A camera-based feedback algorithm monitors and corrects the optoelectronic state of all channels by repeating measurements at four voltages. **b**, **c** On each channel, we balance the measured modulator transmission $\eta$ of each voltage pair ( − , + ) and thus continuously refine alignment with the points of lowest (LO) and highest (HI) extinction. **d** Upon a hardware trigger, we lift the lock and apply desired waveforms to the modulators to control the target systems. After completion, we reinstate the lock. **e** Integrated pulse powers from a sequence of square pulses on one channel. **f** A time trace of the first four microseconds of the same sequence.

Desired waveforms or pulse sequences are applied after a hardware trigger lifts the zero-point lock (Fig. 3d). To characterize the performance of individual channels, we map output light to a fast photodiode and capture time traces. Representative data from periodic square pulses is shown in Fig. 3e, f, demonstrating integrated pulse power deviations below 1%. For optical gates applied to quantum systems, this metric of integrated pulse power indirectly maps to phase accumulated or population driven during a gate pulse. Repeatably

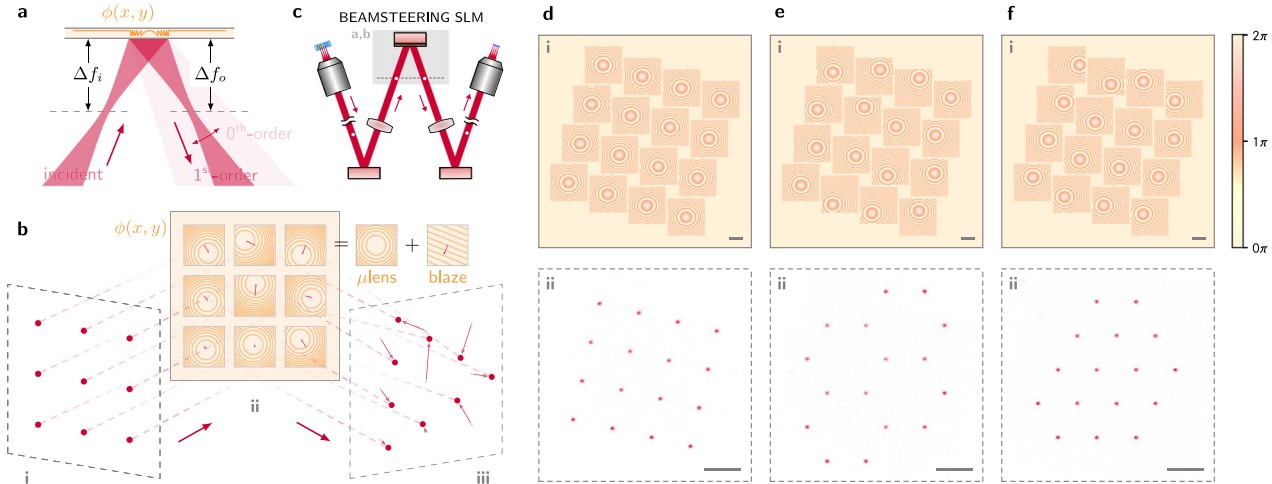

**Fig. 4 | Beamsteering optical channels to target topologies. a** A microlens pattern $\phi(x, y)$ on SLM which is defocused by $\Delta f_i$ is used to refocus incident diverging light to an output imaging plane defocused by $\Delta f_o$ (dashed lines). The small zeroth-order reflection is defocused in this imaging plane, though this can be eliminated via Fourier domain filtering. **b** Adding a steering blaze to each microlens (**ii**) permits reconfiguration of the pattern, in this example from a grid (**i**) to a triangle (**iii**). **c** Our beamsteering optics positioned between the modulators (from left) and the target (to right). Experimental demonstration of steering to example patterns: **d** square (no steering), **e** hexagonal, and **f** triangular lattices.

### Beamshaping

The near-field emission pattern of integrated photonic modulator arrays is fixed by static output couplers, precluding a direct mapping to varying target patterns. Moreover, for atomic control applications, small array fill factors—the ratio of spot diameter to spot pitch—are typically mismatched with densely packed atomic sites[13]. A square array of beams is also topologically mismatched with target patterns of interest in atom optics: ion crystals have increasingly non-uniform pitches towards their ends[14,15], artificial atoms are generally randomly distributed[48,49], and neutral atoms make use of free-form optical trap arrangements for engineered qubit connectivities[5,50,51]. To account for these and other abberation-induced variabilities, our optical control hardware introduces a second LCoS SLM (positioned in a defocused imaging plane) for reconfigurable beamforming on the output.

Dynamic microlenses written to this SLM refocus beams to a shifted output plane (Fig. 4a–c)[52]. Changing the defocusing distance $\Delta f_o$ of the output plane controls lattice fill factor; unity fill is achieved when the beams are each collimated by the microlenses ($\Delta f_o \to \infty$; see "Microlens fill factor conversion" and Supplementary Fig. 2). Linear phase gradients, equivalent to blazed gratings, added to each microlens steer the beams across the output plane to produce spatial spotpatterns matched to desired topologies[5] (Fig. 4d–f). Three-dimensional addressing is also supported by our open-source codebase to enable steering into and out of the plane[50,53]. Pattern uniformity is refined through WGS feedback on the first (fanout) SLM to recover uniformity lost to differing microlens efficiencies and other beampath distortions. We emphasize that each beamspot in each pattern remains an individually-controllable optical degree of freedom which can be modulated with the high speed and precision demonstrated in "Architecture" and "Stabilization".

The degrees of freedom made available by this second SLM are not limited to steering. By honing unique holograms applied to each microlens, we can divide each channel's light into more complicated patterns (Fig. 5a, b). We further enhanced this capability by developing a modified mixed-region-amplitude-freedom (MRAF) algorithm[54] that suppresses higher-order spots—which can introduce coherent crosstalk error as exemplified in Fig. 5ciii—by over 30 dB (Fig. 5diii).

Combining these free-space reconfiguration advances with our integrated modulator array amounts to free-form, high-speed spatio-temporal patterning of many optical degrees of freedom, which unveils regimes of performance for applications ranging from optogenetics[55] to optical ranging[56] and augmented reality[57]. While high-speed optical channels and control electronics will likely continue to come at a premium, these lower-cost, slow degrees of freedom can reshape the spatial basis of action to amplify the overall utility of a system. For instance, Fig. 5a illustrates the projection of alphanumeric characters into free space using our 4 × 4 display, without the need for a much larger grid of channels to finely resolve character structure. Our system can project these characters faster than gigabit ethernet can supply them. For quantum applications, this approach permits general circuit operations on more qubits than available control channels (Fig. 5e, "Quantum circuit factorization"), alleviating system overhead. Finally, the reconfigurability of our architecture also permits pattern healing in the case of non-unity modulator yield, where non-functional channels can be covered by working channels to recover defect-free topologies, albeit with fewer total channels.

### Application

With channel stability and reconfigurability established, we demonstrate the capabilities of our photonic engine using a layer of silicon-vacancy (SiV) color centers in mono-crystalline diamond cooled to 4 K[48]. To characterize this distribution of artificial atoms in space $(x, y)$ and spectrum $(f)$ before applying our photonic engine, we implement parallel-readout photoluminescence excitation (PLE) spectroscopy. Exciting in widefield and collecting on a camera with single-emitter-level sensitivity, we scan the frequency of our incident laser and observe PLE signal on the frequency-detuned phonon sideband (PSB) when individual emitters are resonantly excited. This process reveals an exceptionally narrow SiV spectral distribution for the solid-state—orders of magnitude narrower than implanted samples—made possible by the nature of our sample's growth. Most emitters lie within two peaks split by less than one gigahertz which we attribute to orientation classes of emitters under global strain (Fig. 6a)[58]. We additionally

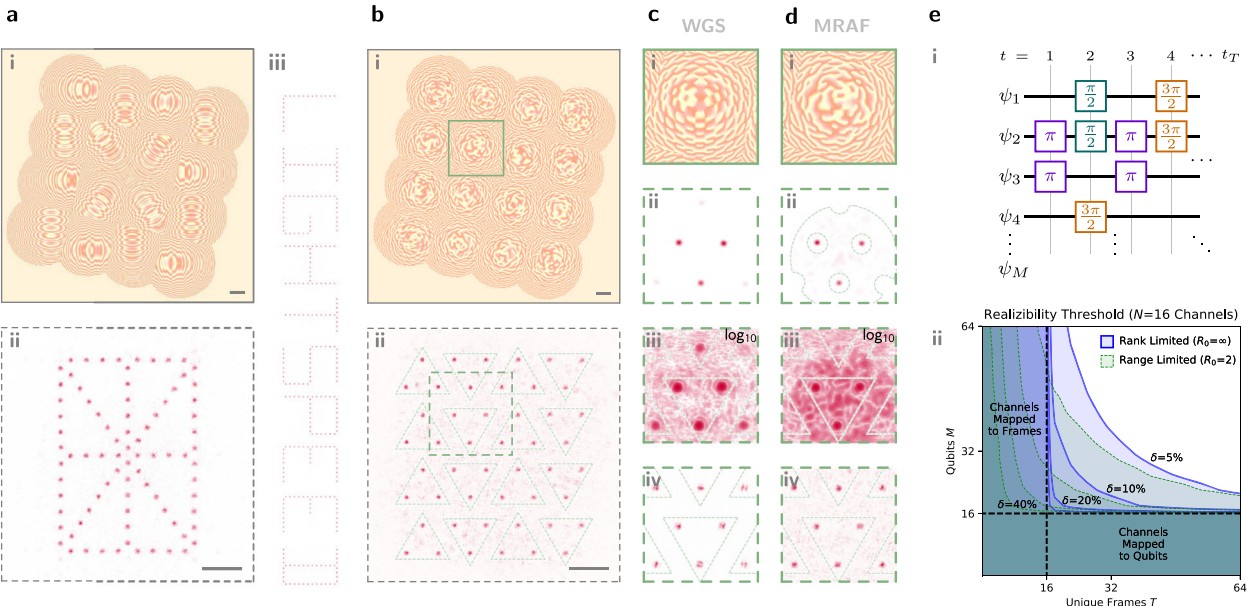

**Fig. 5 | Beamshaping optical channels to target topologies.** Experimental demonstration of shaping: **a** a 16-segment display and (**b**) tiled triplets. Each channel's shaped output can be modulated independently, spelling characters such as "lightspeed" in (**aiii**). **c**, **d** Crosstalk with and without the aid of MRAF. MRAF optimizes to a less symmetric phase pattern (**i**) by allowing power into a noise region (outlined in **dii**). This asymmetry leads to crosstalk suppression (**ii** linear, **iii** logarithmic between $10^{-4}$ and $10^0$) and avoids the interference visible with all channels emitting (**iv**). **e** Such multi-site targeting enables gate sequences (**i**) of unique depth $T$ on a larger number $M$ of qubits $\psi_i$ than there are optical control channels $N$. The curves on (**ii**) represent the realizability threshold or point at which more than half of random circuits with a non-identity gate fraction of $\delta$ are able to be mapped onto hardware. When there are more channels than there are qubits or timesteps ($N \geq T$ or $N \geq M$), there are obvious mappings available, but when the opposite is true there are mathematical limits to realizability due in part to the degree of freedom mismatch or matrix rank deficiency (blue lines; see "Methods" for formalism). Modeling the limited steering range of modulators as $R_0 = 2$ unit cells leads to further restriction (green dashes).

measure most emitters to have linewidths close to the lifetime limit (Fig. 6b). With this knowledge, we set the laser to the center of the denser distribution at $f_0 = 406.70906$ THz (737.11772 nm) and couple power through our photonic engine, steering the output beams to match the positions of sixteen isolated single emitters selected from the field (see "Methods" and Fig. 6c–e).

One of these aligned beams, channel 2, has spatial overlap with a spectrally-detuned second emitter at $f_1 = f_0 + 520$ MHz (Fig. 6f). Using channel 2, we demonstrate spectral addressability on two emitters by performing single-channel PLE, made possible by the GHz-level switching bandwidth of our integrated modulators which exceeds the narrow spread of emitter frequencies in our sample. We set the laser to $f_0 - 4.06$ GHz and frequency shift channel 2 by scanning a tone from 2.5 to 5 GHz, observing the expected peaks at 3.54 and 4.06 GHz corresponding to the two emitters in channel 2 at $f_0$ and $f_1$ (Fig. 6g). In this manner, a sample with wider or otherwise engineered inhomogeneous spectral distribution could be used to extend the number of emitters individually-addressable within a single field of view by a factor corresponding to the number of resolvable spectral sites[49].

In addition to spectral addressing, we implement a 5 µs-long pulse sequence for site-wise spatial addressing. To measure these high speeds with efficiencies inaccessible to our camera, we couple the output emission to a free-space avalanche photodiode (APD), to map the 16 emission spots to the active area of this detector. The pulse sequence consists of three sections, each with 16 pulses on a 100 ns period, demonstrating important concepts in multi-channel quantum optical control: (1) individual emitter addressing, (2) simultaneous addressing of multiple emitters—for example, towards pairwise entanglement generation—and (3) analog tuning of the state of the modulators. Figure 6h–j contains results from channels 0 through 2 and pairs 0–1 through 2–3. More information is displayed in "Methods".

## Discussion

We have proposed and realized an architecture for scalable, visible-wavelength optical programming capable of high-speed control of many optical modes in space, frequency, and time. Indeed, the bandwidth of our lithium niobate modulators enables a sort of true color display where each pixel can spectrally shear light into a desired true optical frequency. Any datarate-limited application in the visible spectrum benefits from our orders-of-magnitude-faster speeds compared to commercial alternatives ("The importance of speed"). Our architecture uniquely reconfigures these fast optical degrees of freedom in space to best match problem-specific spatial bases of control and access problem sizes with complexity beyond the purview of our raw channel count. This extends beyond atomic sites as targets, from probing or controlling patches of living cells with light to projecting data or weights into free-space in the patterns needed to solve machine learning problems at the speed of light. For biology or astronomy, our work could measure and compensate for fast turbulent optical aberration when imaging through adverse environments such as through cells or through the atmosphere. Our work could accelerate any of the many other applications requiring visible light control by enabling faster pattern projection and response. Any datarate-limited application could benefit from greater speed, from mapping augmented realities to sensing through materials with visible transparency windows to ultrafast lithographic patterning of materials.

Immediate scalability is an especially salient feature of these methods and this platform ("The importance of scale"), enabled by integrated fabrication and switching voltages which can be directly synthesized by high-speed CMOS electronics. To this end, the methods described in this work were investigated using system dimensions compatible with chips having much larger channel counts. Only a fraction (<10%) of camera and SLM area ("Microlens scalability") was

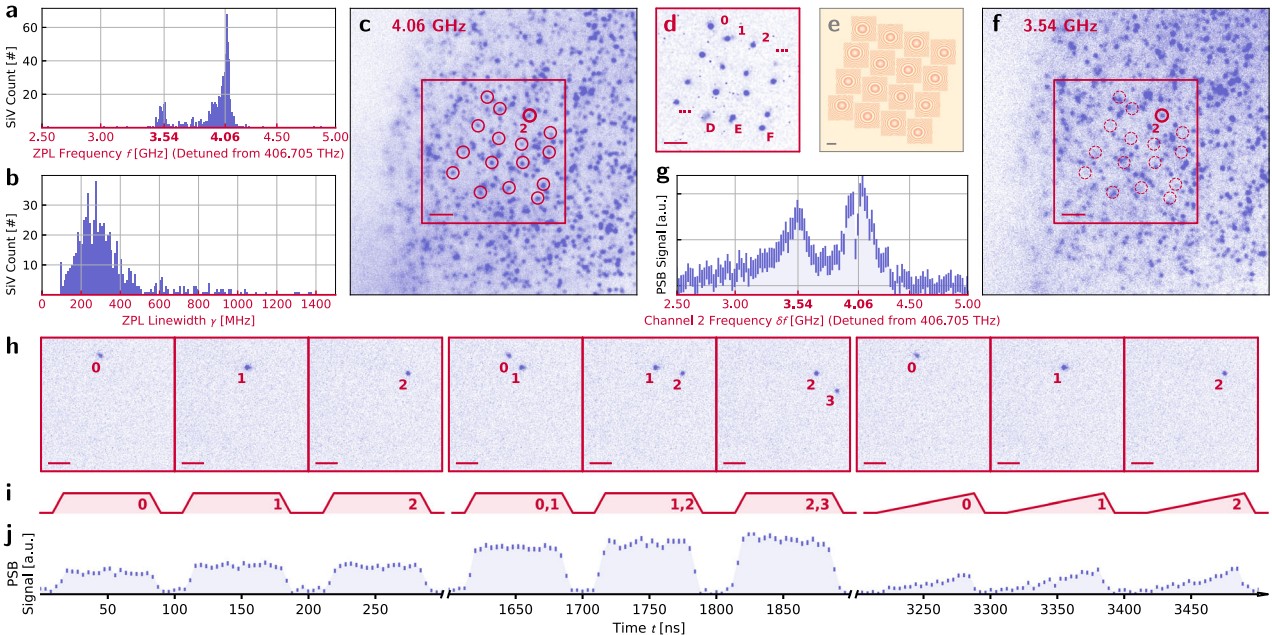

**Fig. 6 | Silicon-vacancy characterization and individual addressing.** Population statistics for 953 emitters within our field of view: **a** the frequency ($f$) distribution of the emitters, revealing two sharp spectral populations, and **b** the measured spectral linewidth distribution. See extended data video 1 for the full three-dimensional data ($xyf$) used to produce these plots. **c** Spatial ($xy$) image of emitters resonant with $f_0 = 406.70906$ THz excitation. We select a subset of 16 emitters (circled) which we **d**, target with our modulator channels (labeled in hexadecimal; 0-F) using **e**, our beamsteering scheme, through the microlens phasemask pictured. **f** Spatial image of emitters resonant with a second color $f_1 = 406.70854$ THz. Channel 2 also has spatial overlap with an emitter at this second spectral plane. **g** A PLE fluorescence scan acquired by shifting the frequency of channel 2, showing peaks corresponding to the two targeted emitters. **h** Sections of a pulse sequence using our multi-channel device targeting different sets of emitters: **i** analog optical waveforms imparted upon the emitters yield, **j** globally-collected fluorescence. Errorbars represent 1-$\sigma$. Scalebars for (**c**, **d**, **f**, and **h**) represent 5 µm. The scalebar for **e** represents 200 µm. Fluorescent spots in d have roughly diffraction-limited waists of ~500 nm. The beams exciting these spots have a slightly larger waist of ~700 nm due to the nature of the linking optics. For these experiments, power of roughly. 5 µW per beam was used, targeting roughly the saturation power of each emitter.

used for the express purpose of field-testing focal lengths, image sizes, and superpixel dimensions suitable for operation with hundreds of channels. Beyond the immediate goal of general quantum simulation with degrees of freedom matching the hundreds of coherent atomic memories in state-of-the-art systems[5,59], these advances would unlock applications of high-speed optical control from brain imaging to astronomy.

## Methods
### The importance of speed
Digital micromirror devices (DMD) or liquid crystal on silicon (LCoS) technologies are commonly used to actuate visible light across a dynamic surface. However, these technologies are fundamentally limited by the speed of mechanical resonances. Commercial devices of these types have stagnated at moderate speeds: $\mathcal{O}(10^4$ Hz) for DMDs and $\mathcal{O}(10^2$ Hz) for LCoS. This work heralds scalable operation of a display-like device at $\mathcal{O}(10^{10}$ Hz), a leap of roughly six orders of magnitude over DMDs.

For quantum computing, from many-body physics investigations to digital gate-based formalisms, this performance leap is essential. Faster individual control on larger sets of atoms could unlock regimes of quantum phases past what is accessible with global control[60]. For digital quantum computing under fault tolerant codes, useful factoring problems are expected to require several days of computation[16] assuming best-case $\mathcal{O}(\mu s)$-duration surface code error correction cycles, each of which includes many gates and a readout step. However, the $\mathcal{O}(> ms)$ cycle times characteristic of state-of-the-art atomic qubit experiments could extend this computation time to decades, assuming low logical error rates could even be maintained over this timescale. That is,

even if logical qubits could be created and stabilized with suitably-low error, a quantum processor still might be too slow to be competitive against classical hardware despite the favorable scaling of asymptotic behavior. In that same manner, a kiloflop processor would be hard-pressed to surpass a yottaflop supercomputer even with the advantage of exponentially-better algorithmic scaling. Quantum computing on atomic species therefore runs the risk of obsolescence without:
- orders-of-magnitude faster cycle times,
- orders-of-magnitude reduced error, or
- orders-of-magnitude better algorithms for error correction or computation.

While both atomic control and atomic readout limit cycle time, this work offers a scalable solution for control to bridge the 4-orders-of-magnitude cycle time gap and operate atoms at the speeds that they are intrinsically capable of.

### The importance of scale
Early demonstrations of atomic computing are made possible by, in part, the mature technology of bulk acousto-optic deflectors and the simplicity of controlling a system with $\mathcal{O}(1)$ capable degrees of freedom[61]. These deflectors act to apply gates or shuttle atoms into gate regions. However, as qubit numbers scale, these degrees of freedom—having a fundamental bandwidth—become saturated in the control they can provide within a given time and a given space, leading to effectively slower gate times. Building more degrees of freedom from bulk components is a logistical challenge that parallels the history of electronic computing. To move beyond discrete components and small numbers of degrees of freedom, the

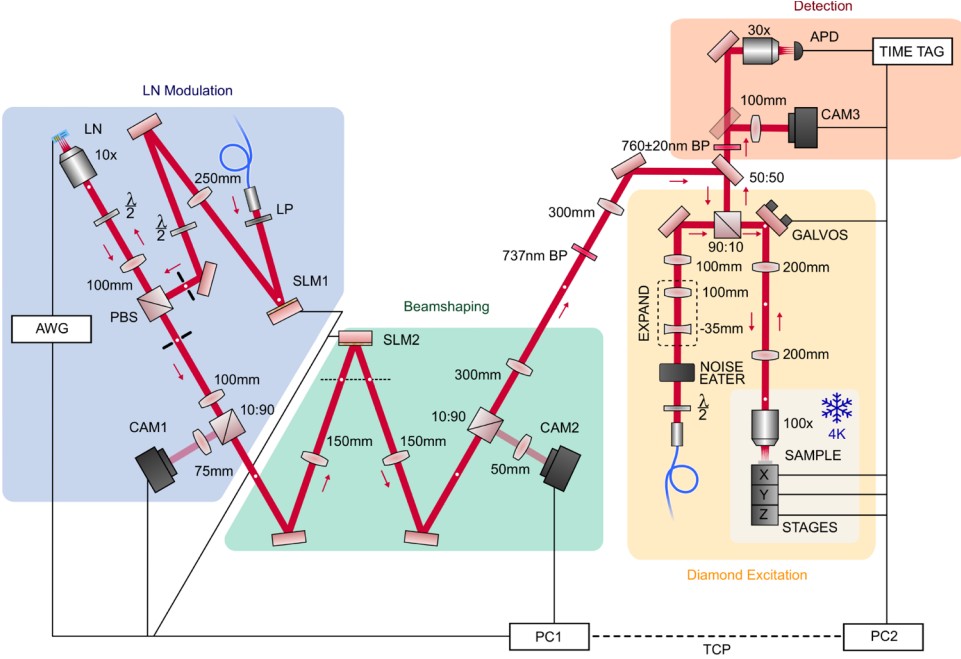

**Fig. 7 | Setup diagram.** Summarizes components detailed in the main text along with Methods sections "Modulator setup" and "Cryostat setup".

semiconductor industry developed technology to print control in the form of miniature, patterned circuitry. Our architecture is the analog of this for quantum optical control: the technology to print many optical control channels with high bandwidth and have them work in a greater system to enable $\mathcal{O}(100\,\text{ns})$ gate times. In particular, the CMOS-compatible voltages enabled by TFLN offer immediate opportunities for integration with control electronics. In this vision of distributed control, managing millions of qubits is not an impossibility, but rather the challenge of engineering a scalable unit cell, where each printable cell contains the electronic logic and optical control to sustain a qubit or set of qubits, with light then projected and demagnified to the qubits by free-space optics. Notably, the demagnification of the optical train permits a large control unit cell pitch despite the small pitch $\mathcal{O}(5\,\mu\text{m})$ used for atoms, avoiding the density, interconnection, and cooling problems that have plagued other qubit modalities. With the addition of local electronics, each ultrafast pixel of our architecture can fit inside a $200 \times 200\,\mu\text{m}$ unit cell, which is approximately the size of the pixels used in commercial HD laptop displays (containing a million pixels).

## Modulator setup

Figure 7 summarizes our optical setup. We collimate light from a polarization-maintaining fiber (Thorlabs PM780-HP) to a free-space beam with cm-scale diameter (Thorlabs ASL10142M-B; 79 mm asphere). White light (Thorlabs MWWHF2) is flipped into the path for diagnostic images. The beam reflects off the fanout SLM (SLM1; Thorlabs Exulus-HD3). The plane of the SLM is scaled and mapped to the back aperture of the main objective (Zeiss Plan-NeoFluar, 10×, 0.3 NA) via a $4f$ (250 mm and 100 mm achromats), passing through the $s$ port of a polarizing beamsplitter (PBS; Thorlabs PBS252). The hologram produced at the plane of the chip by SLM1 couples through each modulator channel and out the $p$ port of the PBS. A second $4f$ (100 mm achromats), which shares a 100 mm lens with the first $4f$, couples light to an intermediate Fourier plane. From there, the light is partially reflected to a first pickoff camera (CAM1; Thorlabs Quantalux; 75 mm imaging achromat) to monitor the state of our channels. The remaining light propagates further to the beamshaping setup. The beamshaping SLM (SLM2; Santec SLM-200-01-0002-02) is positioned at a

defocused imaging plane between two lenses (150 mm achromat). A second pickoff camera (CAM2; Thorlabs Zelux; 50 mm imaging achromat) monitors the state of beamshaping. From there, the light propagates to the cryostat setup described in "Cryostat setup".

This setup, including the modulators and beamshaping optics, are controlled by a computer (PC1) with Python code. Experiments that do not include color center addressing are controlled directly from this computer. 737 nm light is sourced from a Ti:Saph laser (MSquared SolsTiS) stabilized with a wavemeter (HighFinesse WS7; accuracy 60 GHz, precision 2 MHz). For data taken at 780 nm, we use an external cavity diode laser (New Focus Velocity TLB-6712). Each modulator is connected to an arbitrary waveform generator (AWG) channel sourced from four 4-channel cards (4× Spectrum Instruments M2p.6566-x4, 16-bit, 125 Ms/s, 70 MHz). For frequency shifting experiments, a faster AWG with only two channels (Tektronix AWG70002A, 25 Gs/s, 10 GHz) is used with a broadband amplifier (Centellax OA4MVM, 30 dB, 45 GHz) to produce chirped sweeps. Fast visible-wavelength photodiodes are used for high-speed measurements: for time traces (Melno Systems FPD510, 200 MHz) and bandwidth measurements (Newport 818-BB-45AF, 10 GHz). An oscilloscope (Agilent Infiniium DSO81004A, 10 GHz, 40 Gs/s) and a microwave vector network analyzer (Keysight N5224A PNA, 10 MHz to 43.5 GHz) are used to record signal from these photodiodes.

## Chip fabrication

Our chip was fabricated using a pre-commercial multi-project wafer (MPW) foundry service by CSEM. The fabrication process flow begins with thinning down a $x$-cut lithium niobate on insulator (LNOI) thin film (NanoLN) from 600 nm thickness to 200 nm using blanket etching (Fig. 8a, b). The 200 nm thickness is optimized for performance at wavelengths around 780 nm. The resulting 200 nm LNOI thin-film is patterned using an HSQ mask via electron-beam lithography and etched a further 100 nm to yield low-loss optical TFLN waveguides with a sidewall slant ~30° from normal (Fig. 8c). For active electro-optic structures, 500 nm gold electrodes are patterned via liftoff (Fig. 8d). We use 400 nm wide traces as standard single mode routing waveguides. These are tapered out slightly to 500 nm inside the phase shifters to reduce propagation loss. Phase shifters use a $3\,\mu\text{m}$ gap

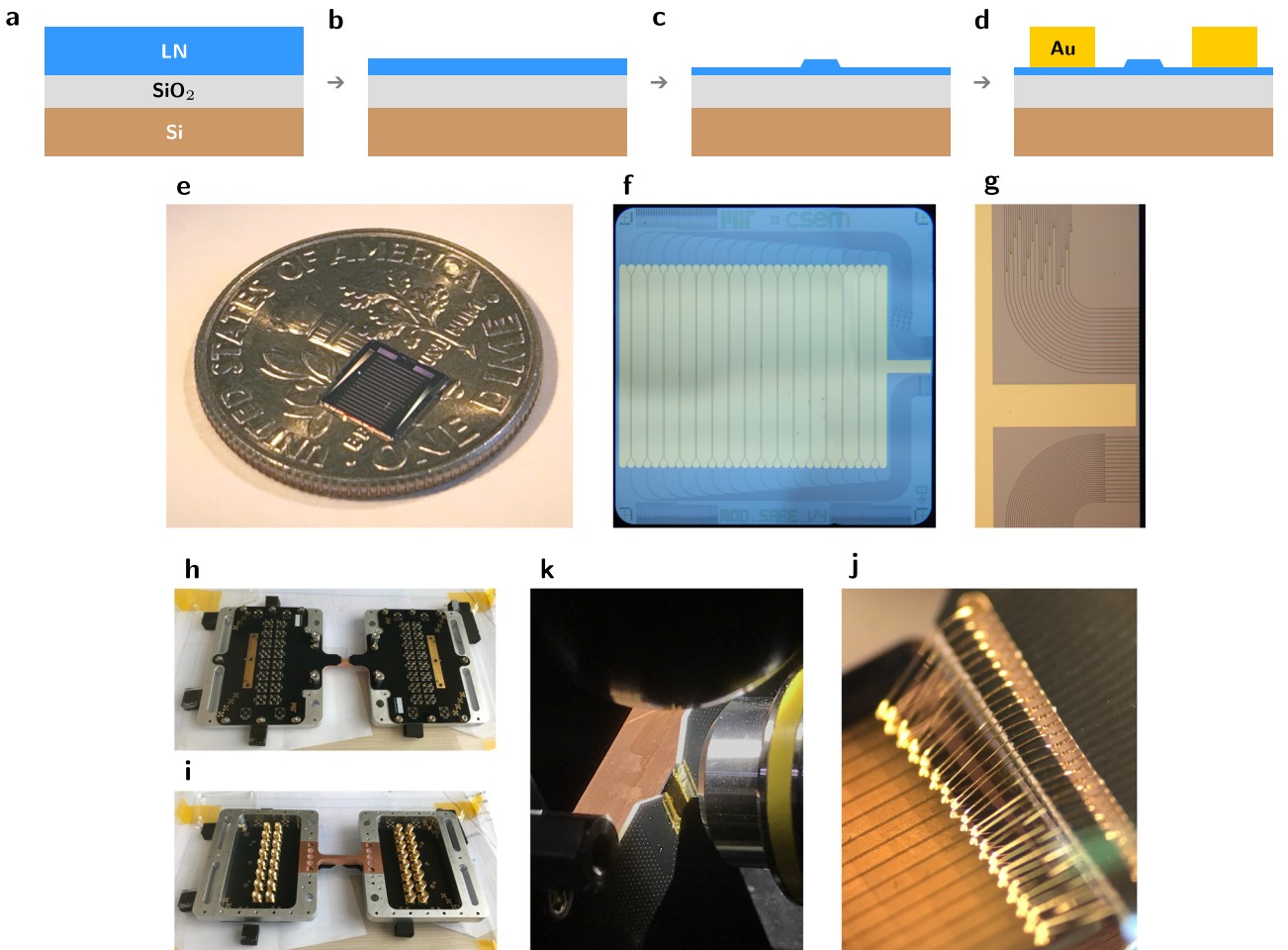

**Fig. 8 | Chip fabrication, images, and packaging. a**–**d** Fabrication follows steps detailed in "Chip fabrication". **e** Our 5 mm × 5 mm chip was imaged on top of a dime for scale and **f** under an optical microscope. **g** Zoom upon the grating coupling region. **h**, **i** The top and bottom of our chassis before chip placement. The chip sits at the center of copper beam in (**h**). **j** A wirebonded chip positioned in front of the objective. **k** Zoom upon an array of wirebonds.

between electrodes. Waveguides are tapered down to 200 nm in directional couplers to increase the mode overlap between the two directional coupler arms.

### Chip layout

The photonic engine architecture described in the main text is fabricated on a 5 mm × 5 mm chip, consisting of a bank of sixteen MZIs with cross ports routed to banks of input and output grating couplers (Fig. 8f). The opposing two cross ports of each 2 × 2 MZI are routed to edge couplers via waveguides threaded inbetween the grating waveguides (Fig. 8g). Our system could be operated with these edge couplers instead of the grating couplers. In fact, our packaging was designed with this feature in mind, where the design of the chassis is compatible with a 90° rotation to expose the edge facet to objective imaging (Fig. 8h–j). Edge-coupled operation is expected to perform with efficiencies closer to 2 dB channel insertion loss, given compatible optics and appropriate mode-matching. Quantum systems with 1D topologies, such as linear ion crystals, are likely well suited to edge-coupled operation. However, in this work, we chose to focus on grating-coupled operations for scalability. Assembling a million control channels in 1D requires a thousand times more lateral distance than assembling a thousand-by-thousand 2D array. It is thus no coincidence that equipment for beam monitoring (cameras, e.g. Fig. 3) and beam steering (SLMs, e.g. Figs. 4, 5) offering millions of readout or control channels occur commercially only in two-dimensional patterns.

### Chip packaging

The chip is glued with thermal epoxy (Arctic Silver) to a custom copper mount. The copper mount is secured to custom aluminum parts, insulated with a layer of kapton tape and using teflon screws to enhance thermal isolation. The copper mount is thermally stabilized with a thermoelectric cooler and temperature controller (Arroyo 5300 series). The aluminum parts act as a mounting plate and support structure for custom FR4 PCBs which interpose sixteen SMA ports with the ground-signal-ground wirebond pads used to interface with our chip. Together, these elements make up the chassis (Fig. 8h–i). Chip and PCB pads are connected manually with a gold ball wirebonder with unity yield (Fig. 8j) allowing operation of every modulator. Our bandwidth is likely limited by the roughly 50 mm long traces on our lossy FR4 PCB, along with contributions from wirebond length and compactness.

### Vertical coupler design and efficiency

Our grating couplers are designed to couple light from a single-mode waveguide to a vertical Gaussian beam with a 0.12 numerical aperture (NA) at 780 nm. Vertical coupling is important to remain within the NA of the normally-incident objective. We used gradient-based optimization[62] in the longitudinal direction (i.e. waveguide propagation direction) and shaped the transverse focus to that of the longitudinal direction using analytic assumptions of Gaussian phasefronts. We measure a slight ellipticity to the beam in waist and focus, corresponding to mismatch between the transverse and longitudinal

directions, though this is compensated with the beamshaping microlenses. Metal guards (visible as U shapes in Fig. 2di) are fabricated around the output gratings with the intent of reducing crosstalk scatter to neighboring outputs. The device (and, implicitly, grating) efficiencies reported in the text are measured as the ratio between the power at the input ($s$) port of the PBS and the power at the output ($p$) port of the PBS.

We engineered degrees of freedom in our setup to account for deviations from vertical grating coupling. The mirror on the excitation path immediately before the polarizing beamsplitter is designed to be near an imaging plane (see white dot), such that changing the angle of this mirror tunes the angle of excitation at the sample's imaging plane to optimally couple the hologram into the gratings. At 780 nm, we find that the optimal angle of excitation is close to normal and measure roughly 20 dB device insertion loss. At 737 nm, we measure closer to 40 dB device insertion loss. We attribute this additionally loss to a non-vertical grating coupling angle at this non-design wavelength, finding evidence in the wavefront calibration for 737 nm, where the angle producing optimal coupling also causes significant amplitude distribution clipping (Supplementary Fig. 1b). We interpret this as clipping on the edge of the objective's NA.

## Chip improvements
Going forward, we see a number of paths to improve the efficiency, extinction, and compactness of our devices beyond these first-iteration designs.

The efficiency of our system was dominated by the insertion loss of our grating couplers. These gratings act as integrated Bragg-like reflectors to scatter light into a free-space mode, but are sensitive to small variations in feature sizes and couple as designed over a limited wavelength range. Mature and highly-controlled photonic processes, such as those in silicon photonics, are able to reach dB-level losses per grating[63]. In thin film lithium niobate, controls are arguably limited by non-uniform film thicknesses, especially compared with the maturity of silicon-on-insulator, and lateral imprecision in fabrication from variations in resist exposure, resist development, or etching conditions. These issues could be circumvented by precharacterization of film thickness or by isolating processes on dedicated tool lines to improve process reliability, as is done in silicon photonics. As an alternative approach, total-internal-reflection-based polymer mirrors could be fabricated to efficiently direct edge-coupled light upward, eliminating coupling loss and narrow-band operation as concerns[64,65].

Device extinction ratio, important for high fidelity quantum control, is likely limited by differences in splitting ratio between the two directional couplers which make up each MZI. Avenues for improving extinction include optimizing coupler parameters and topology for increased tolerance to fabrication imperfections, or tuning couplers to recover performance. At the cost of additional complexity, multiple MZIs could be cascaded per channel to reach exponentially-higher levels of extinction.

The compactness of current uncladded devices is constrained by a single in-plane layer of metal, which forces waveguides to be routed around large wirebond pads. An additional out-of-plane layer of metal will allow us to approach closer to the limit of TFLN MZI compactness and realize substantially more channels. While the footprint of these millimeters-long MZIs is much larger than that of resonant devices[28], channel count is ultimately limited by the length of a chip's perimeter in wirebonded control architectures—as the number of bondable electrical pads scales with perimeter—or the area of electrical drivers in a fully-integrated approach.

## Fanout wavefront calibration
The generation of diffraction-limited holograms depends on accurate characterization and compensation of beampath aberrations. We automatically measure the source laser's phase and amplitude distributions at the plane of the SLM via imaging domain interference of light diffracted from SLM superpixel clusters[66]. Gerchberg-Saxton (GS)-type algorithms then use this measured amplitude distribution when numerically generating phase profiles, enabling greater holographic accuracy. We find that the amplitude distribution becomes significantly clipped as the reference position approaches the edge of the field of view. To compensate for this, we purposefully overclip the SLM with a physical iris such that the amplitude distribution is equally clipped regardless of the reference position (clipping oval from the iris is visible on the amplitude distribution pictured in Supplementary Fig. 1a). This allows us to generate more accurate holograms across the region near the edge of the field of view where the input grating array is located.

## Imaging domain calibration
We locate the positions of input and output gratings in imaging domain $x$-space manually using a reference image previously taken with white light. The positions of the output gratings are used to center pixel integration regions about each output spot, which we use to determine the output power of each channel. We also calibrate the coordinate transform between SLM $k$-space and the imaging domain of the chip. This amounts to generating a grid pattern on the SLM and fitting the result to an affine transformation.

## Fanout hologram generation
For this section, a sawtooth electrical signal with peak-to-peak amplitude of $2V_\pi$ at a frequency of 100 cycles per camera frame is applied to every modulator. This tone averages out fluctuations in device transmission due to the electrical degree of freedom, and isolates the problem of optimizing fanout coupling. The inverse of the calibration coordinate transformation and $x$-space input grating positions found in "Imaging domain calibration" are used to target positions in SLM $k$-space during GS fanout hologram optimization. This first guess hologram is rarely perfectly aligned, which we attribute to chromatic aberration between the white light image and the target wavelength along with imperfections in our coordinate transform fitting. To correct this mismatch between guess and true $k$-vectors, we add a global steering blaze to the SLM and scan the position of the hologram across the gratings, measuring the coupling through the output gratings via camera readout. We iteratively correct the $k$-vector guess for each channel by adding the blaze $k$-vector that produced maximal coupling in the previous iteration. Four iterations yield alignment below measurement noise.

With the $k$-space map corrected, we use WGS to compensate for remaining pointing or device transmission errors. For the data presented in this work, we optimize the uniformity of integrated spot power, though other figures of merit such as spot amplitude uniformity are equally applicable for weighted optimization. We believe that current uniformity is limited by small mechanical vibrations between the objective and chip causing pointing errors between the fanout hologram and input gratings. This effect can be mitigated in future iterations by directly securing the objective to the chassis that supports the chip, rather than separately securing them upon the same optical breadboard. Without any optical feedback or temperature stabilization, our system and holograms remain stable to within 10% of baseline coupling for 24 h.

## Closed-loop zero-point stabilization
We preload the four-voltage locking sequence and desired pulse sequence to the AWGs. The camera used to measure the state of each modulator at these voltages is triggered via a digital AWG line to maintain synchronization. Updates to setpoints are applied to the amplitude and offset voltage parameters of the AWGs, which scale the

unitless digital waveforms loaded to memory, avoiding time-consuming waveform memory rewriting. We prioritize (via CPU-based analysis) updating channels that have the largest absolute error between the current setpoint and the measured target value, as updating these parameters is still slow (to the point that we can only update two parameters per camera frame). Hardware without such update latency limitations can be used to eliminate this issue.

## Microlens fill factor conversion

We define the fill factors $\eta_i$ and $\eta_o$ of the input and output beams as the ratios between the $1/e^2$ beam diameters $2w_i$ and $2w_o$ and pitch $\Gamma$ of spots. For filled microlenses ($2w(z) = 2w(\Delta f) = \Gamma$) and an approximately diffraction-limited system, we estimate the defocusing distances $\Delta f_{i|o}$ between the SLM and the input and output planes to follow the relation:

$$\Gamma = 2w_{i|o} \sqrt{1 + \left(\frac{\Delta f_{i|o}\lambda}{\pi w_{i|o}^2}\right)^2}, \tag{1}$$

where $z_R^{i|o} = \pi w_{i|o}^2/\lambda$ is the Rayleigh range. While the input defocusing distance $\Delta f_i$ is fixed by the choice of pitch $\Gamma$ and the fill factor $\eta_i$ of our system, we can engineer $\Delta f_o$ to target a desired $\eta_o$ (Supplementary Fig. 2) according to:

$$\Delta f_o(\eta_o) = \frac{\pi}{4}\frac{\Gamma^2\eta_o}{\lambda}\sqrt{1 - \eta_o^2}. \tag{2}$$

Unity fill factor can be achieved when the output beams are collimated (Supplementary Fig. 2d).

## Microlens steering range

There are two main factors that limit steering range: (i) the SLM's diffraction efficiency versus steered angle, and (ii) the extent to which steering can be realized through high magnification objectives.

For commercial LCoS SLMs, diffraction efficiency degrades as target blaze gradients approach the point where they can no longer be resolved by finite pixel size. The half-angular SLM steering bandwidth $\theta_{max}^{diff}$ is usually on the order of two degrees (the Bragg condition at 780 nm for a sawtooth blaze with a pitch of three 8 μm pixels).

Steering angles in the domain of the SLM are mapped to angles at the domain of the target, multiplied by a factor corresponding to the objective magnifcation $M$. For the high ($M \sim 50\times$) objectives found in atomics experiments and given an appropriate imaging lens, a steering angle of two degrees at the SLM will not propagate through the objective as $M \times 2° = 100° > 90°$ exceeds the NA of free space. A shallower angle may also clip upon the NA of the objective, bounded by $\theta_{max}^{obj} = \theta_{NA}/M$. This limit could be completely negated with a third SLM used to reorient the angle of the beams to vertical incidence after they are spatially steered by the second SLM.

These bounds on steering angle pose a limit on $r$, the steering range, and $R$, the steering range normalized to pitch $\Gamma$:

$$R = \frac{r}{\Gamma} < \frac{\theta_{max}\Delta f_o}{\Gamma} = \theta_{max}\frac{\pi}{4}\frac{\Gamma\eta_0}{\lambda}\sqrt{1 - \eta_0^2}. \tag{3}$$

In our system, with $\Gamma \sim 0.65$ mm and $\lambda \sim 780$ nm and using $\theta_{max} = \theta_{max}^{diff}$, this evaluates to roughly $R \sim 20\eta_o$. For small $\eta_{i|o}$, the phase gradient of the parabolic focusing phase can also be large enough to locally exceed $\theta_{max}^{diff}$ on the edges of the lens, limiting $R$ further. For the data presented in Fig. 5, we use $\eta_o \sim 0.05$. We separately measure $R$ for this $\eta_o$, finding that steering within $R \sim 1$ maintains efficiencies over 80%, in agreement with $R \sim 20\eta_0$ (Supplementary Fig. 3).

Though we operate the beamshaping SLM at a roughly five degree reflection angle, this does not mitigate performance. That is, focal detunings resulting from this slant (e.g. at the edges of the microlens grid) do not exceed the Rayleigh length of the beams, for the input and output parameters that we consider. Aberrations from these small focal detunings are corrected on a microlens-by-microlens basis via automated routines.

## Microlens pattern steering

For the topologies displayed in Fig. 4c–f, we automatically steer beams to a target pattern defined in the beamshaping camera's (CAM2's) basis. Channels are matched to target points by framing the task as a linear sum assignment problem. Using knowledge of the focal lengths and defocusing distances, we analytically calculate and apply the blaze necessary to steer each channel to the target position. Measuring the positional error between the steered position and the target, we iteratively apply this process until the channels are sufficiently aligned. Four iterations yield satisfactory alignment. We do not observe significant spot profile degradation with steering. As steering angle is tuned over a "1 unit cell" distance, the spots have ~9% peak-to-peak and 2% standard deviation beam radius variation across all sixteen imaged spots.

## Microlens pattern shaping

Similar methods apply for shaping patterns from each channel, with linear sum assignment to pattern centroids, though this non-analytic steering requires further calibration to achieve better numerical hologram optimization. We apply the same amplitude and $k$-space calibration techniques as in "Fanout wavefront calibration" and "Imaging domain calibration", except now done simultaneously on each of the microlenses, with differences to account for the smaller $x$- and $k$-spaces resulting from each microlens' footprint as a small fraction of the total SLM area. Notably, the farfield of each microlens is chosen to be the focused farfield at the output imaging plane, so this calibration occurs under a sum with an analytic lens. The $k$-space calibration on each focused microlens is done serially by scanning all the beams, instead of in one shot by projecting an array of spots, due to the Nyquist limits of the smaller $k$-space.

For MRAF holograms, automated routines construct a noise region at the center of each $k$-space, with subregions excluded around the site of each targeted spot. MRAF acts via an attenuative term in the WGS loop (Supplementary Fig. 7c) which avoids injecting power into neighboring spots.

## Quantum circuit factorization

Our architecture's unique combination of fast and slow degrees of freedom—combined with channel-wise output power splitting—enhances quantum circuit compilation. A quantum circuit, for simplicity, is assumed to be a series of $T$ single-qubit gates on $M$ qubits. We sample these gates from a set of four: $\{I, X^{1/2}, X, X^{3/2}\}$, where $X$ is a $\pi$ rotation around the $\hat{x}$ axis. This circuit can be represented as a $M \times T$ matrix $\Theta$ with entries $\{0, 1, 2, 3\}$ corresponding to multiples of $\pi/2$. We define the density $\delta$ of $\Theta$ to be the fraction of elements that are nonzero (i.e. not the identity). It is natural to expect that a more sparse set of circuits results in relaxed requirements on the control, in general.

To synthesize these gates, we have hardware with fast and slow degrees of freedom. The fast degrees of freedom correspond to $N$ optical modulator channels and can be represented by a $N \times T$ matrix $\Phi$ corresponding to each channel power setpoint at each timebin. The slow degrees of freedom correspond to (effectively static) fanout from each modulator channel to our qubits, a $M \times N$ matrix $\Gamma$. The modulator matrix $\Phi$ is unbounded, but should be positive to always accumulate phase. Large $\Phi$ corresponds to making the time bins larger or injecting

**a**

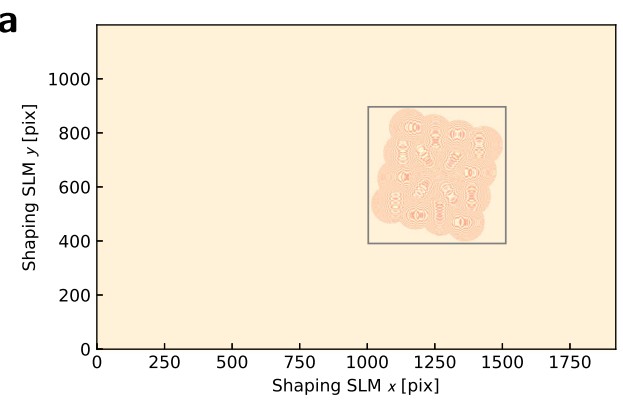

**b**

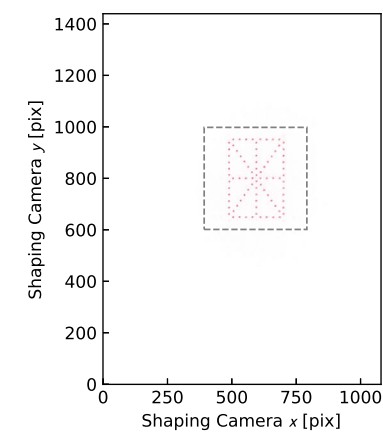

**Fig. 9 | Testing scaled dimensions.** The full fields of view of Fig. 5a, consisting of (**a**), the shaping SLM and **b**, the shaping camera, show that only a small fraction of the area (<10%) is used for testing, implying immediate scalability to larger (by >10×) channel counts. Squares mark the zoom regions of Fig. 5ai, aii. Similar dimensions are used for Fig. 4 also.

more power into the system. Then, control $\Theta$ is achieved as a simple matrix multiplication:

$$\Theta = \Gamma\Phi. \tag{4}$$

Compiling a quantum circuit on this hardware amounts to finding a suitable factorization of $\Theta$ which stays within the limits of hardware (Supplementary Fig. 6). We have one additional constraint on $\Gamma$: some qubits are out of range of our slow fanout degrees of freedom. This corresponds to when qubits are spatially outside the range of the Fourier domain of each fanout degree of freedom. We can model this by forcing elements $\Gamma_{mn}$ to be zero when out of range, or when $|\vec{\psi}_m - \vec{\phi}_n| > R_0$, where $\vec{\psi}_m$ is the position of the $m$th qubit, $\vec{\phi}_n$ is the position of the center of the $n$th modulator's Fourier space, and $R_0$ is the operating range of the microlens degrees of freedom.

    We developed a combinatorial search algorithm that looks for temporal or spatial symmetry within a circuit to find an effective factorization. Quantum codes often have such symmetry, with repeating patterns in time and space, so the random circuit-set explored acts as a lower bound on performance. When $M$ or $T$ is greater than $N$, there are cases where a high-rank $\Theta$ with rank($\Theta$) >$N$ cannot be described as the product of lower-rank matrices rank($\Gamma$), rank($\Phi$)≤$N$. This rank limit afflicts a portion of random $\Theta$ matrices which increases with non-identity gate density $\delta$. The proportion of factorable $\Theta$ is reported in Fig. 5eii for two cases: a range-limited case where $R_0$ is small enough to influence factoring and a rank-limited case where $R_0$ is ignored and the underlying factorability of $\Theta$ is exposed.

**Microlens scalability**

The 4 × 4 microlens arrays used in this work comprise only 6% of the area of the shaping SLM (Fig. 9a), demonstrating the feasibility of scaling channel counts by more than an order of magnitude. Figure 10 confirms this assertion by: 1) examining the relative microlens efficiency under experimental aberration, and 2) plotting the smallest feasible microlens size under different conditions. Extrapolating further, we find that the number of microlenses which can be supported by the SLM corresponds roughly to the number of Nyquist-limited spots with a similar fill factor $N_x \sim N_x^{\text{pix}} \eta_o$ one could create directly from using the full field of view of the SLM.

    From the perspective of atoms in trapped optical tweezers, this invariant means that the amount of control possible under our architecture matches the number of tweezers which can be created with similar SLM hardware. More control is possible by tiling SLMs or by taking advantage of the commercial trend towards larger pixel count SLMs, with 4 K resolutions already available. This analysis also reveals

interesting tradeoffs. For large microlens patterns with corresponding long focal length and phase pattern with fewer discontinuities, blurring can improve performance as phase is smoothed to closer approximate an ideal parabola from a discrete distribution. However, in the case of strong steering, the smoothing instead acts negatively on the sawtooth blaze of the uncentered parabola.

**Cryostat setup**

Figure 7 summarizes our optical setup. Modulated and beamshaped light passes through a 4$f$ (300 mm achromats) from our modulator setup, passing through a cleanup filter (ZPL BP; Semrock BrightLine 740/13). This merges with the excitation path of our cryostat setup with a removable 50:50 pellicle beamsplitter (Thorlabs BP145B2). Galvos normally used for scanning confocal microscopy are used in this work for fine adjustments. These galvos are mapped by a 4$f$ (200 mm achromats) to the back aperture of a cryo-optic objective (Zeiss 100×, 0.95 NA) inside a 4 K cryostat (Montana Instruments Cyrostation s50). Our diamond sample is positioned under the objective with peizo slip-stick stages (Attocube). Emission light is collected back through the same path, to the other port of the pellicle. Excitation light is filtered out, leaving the phonon sideband (PSB BP; Semrock Brightline 775/46). From there, we collect on either a electron-multiplying charge-coupled-device (EMCCD) camera (CAM3; Photometrics Cascade 1 K; 100 mm imaging achromat) or to an APD (Excelitas Technologies SPCM-AQRH-14) with time-tagged signal (Swabian Instruments Time Tagger 20). For widefield characterization data collected with the camera, we remove the pellicle and use a separate excitation port input on the 10 sides of a 90:10 beamsplitter. For this port, light is focused at the galvos (100 mm lens singlet), and thus−via the 4$f$−focused at the back aperture of the objective. This focused beam in the Fourier domain of the objective maps to a widefield beam in the imaging domain. A beam expander (−35 mm singlet, 100 mm singlet) increases the NA of the beam at the back aperture to fill the field of view in the imaging domain. A compact noise eater (Thorlabs NEL03A) is used to stabilize power fluctuations occurring during laser wavelength sweeping. Diamond experiments are controlled from a computer (PC2) which interfaces with the modulator computer (PC1) via an ethernet (TCP) link. A mixture of MATLAB and Python code is used for these experiments.

**Silicon-vacancy sample**

Our diamond sample was produced by chemical vapor deposition (CVD) overgrowth on a low-strain mono-crystalline substrate (New Diamond Technologies). Silicon is incorporated during the CVD

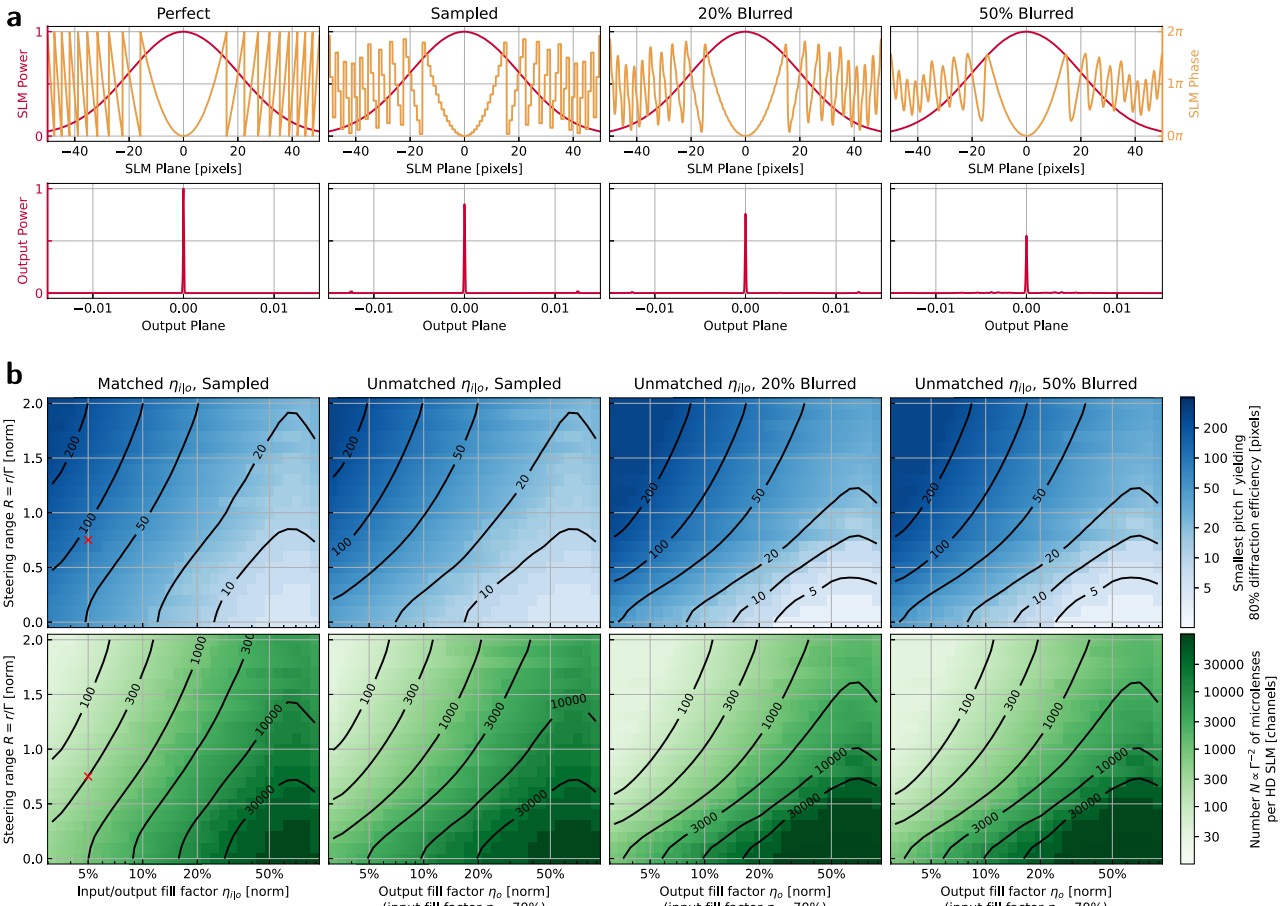

**Fig. 10 | Microlens scalability. a** The nearfield phase and amplitude upon the SLM plane are related to the farfield output by a Fourier transform. While a perfect parabolic lens results in a perfect focused farfield spot, the incorporation of various non-idealities reduces the power efficiency of coupling. Treated here are the most relevant errors: the discrete sampling of phase from each SLM pixel, and the blurring of phase between pixels expected to occur on hardware. Here, this effect is modeled as a symmetric exponential $e^{-|x|/\alpha}$ blur with characteristic length $\alpha$ corresponding to a percentage of the pixel width. **b** From these simulations, we can extrapolate the performance of an array of microlenses for various steering ranges and fill factors. Larger microlens size generally monotonically corresponds to greater efficiency, so for each case we mark the smallest size of the microlens that is more than 80% efficient. Reciprocal to this is the related figure of merit of microlens number $N = A_{HD}/l^2$ which can be packed onto an HD SLM with pixel area $A_{HD}$. The parameters used in this work are indicated by the red crosses.

process, yielding SiV color centers in the overgrown layer. CVD overgrowth has previously been shown to generate narrow SiV inhomogeneous distributions similar to what we observe in this sample[67,68]. Following overgrowth, the sample was cleaned by boiling in a 1:1:1 mixture of nitric, sulfuric, and perchloric acids for 1 h[69], cleaned in a 3:1 sulfuric acid:hydrogen peroxide piranha solution for 5 min, rinsed in a solvent bath, and annealed at 1200 °C in ultra-high vacuum. To remove any graphite formed during annealing, the sample was again cleaned in tri-acid and piranha using the same processes.

### Channel-emitter alignment
We adopt techniques from scanning microscopy, and use each microlens to scan each channel over a region of diamond (Supplementary Fig. 7a). For each channel, we select a point in each scan corresponding to an isolated emitter. Then, we iteratively optimize the blaze and focus of each microlens, maximizing the camera signal corresponding to the chosen isolated emitter (Supplementary Fig. 7b).

### Electrical breakdown damage
After aligning our modulator channels to a set of sixteen SiV emitters, we proceeded to begin gathering final fluorescence data. Despite working consistently for months beforehand, nine of our channels

stopped coupling light during this acquisition (channels 4 through B and channels E through F). We attribute this channel loss event to an untimely Windows Update producing an over-voltage state on the PCIe AWG cards which were active at the time, thus exceeding the electrical breakdown field of the air gap between the uncladded electrodes. The resulting sparking across the gap accounts for the delaminated electrodes and charred waveguides which we observe on afflicted channels (Supplementary Fig. 8). As a result, the spatial addressing demonstration detailed in Fig. 6 uses only the remaining modulator channels.

Breakdown can be nucleated by small sharp features which locally enhance electric field[70]. The stochastic presence of such sharp features on each channel is a potential cause of the partial, rather than complete, destruction of our channels. Future work using a silicon dioxide[71] cladding will eliminate this problem as lithium niobate[72], a stronger dielectric than air by more than an order of magnitude[73], becomes the limiting material for electrical breakdown. While this comparison is not strictly valid over the relevant electrodes gap sizes—due to differing breakdown mechanisms between solids and gasses along with raised breakdown thresholds for small gaps[70]—the cited works illustrate the relative trends for these materials. In cases where cladding is impractical, external overvoltage protection or alternative electronics could be used to prevent damage should a power outage or automated update occur.

## Spectral and spatial addressing

The spectral plot Fig. 6g is acquired by chirping channel 2 at a frequency $\delta f$ using our 25 Gs/s AWG and collecting signal on our EMCCD camera. Error is estimated from Poisson statistics. Data are acquired with 10 MHz steps, then binned to 20 MHz steps for visibility. The choice of our sweep from 2.5 to 5 GHz stays within the bandwidth of our AWG and amplifier while avoiding spurious signal from higher-order sidebands ($2\delta f$).

Our pulse sequence, as described in the main text, consists of a series of 80 ns pulses inside 100 ns bins. Figure 6h illustrates the spatial state of our modulators with camera images corresponding to a set of pulses, though these images are integrated for 5 s, instead of the 80 ns in the pulse sequence. Figure 6j was collected via repeating our pulse sequence over 20 min of integration. Error is estimated from Poisson statistics. The risetime slopes in Fig. 6i, j represent transitions from setpoint to setpoint of our 125 MS/s AWGs (8 ns per setpoint). This risetime is nevertheless competitive with commercial AOMs.

## Data availability

The data generated in this study have been deposited in the Zenodo database under accession code https://doi.org/10.5281/zenodo.14225442.

## Code availability

Available on request to I.C. SLM-related routines are public on GitHub under the slmsuite package.

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

## Acknowledgements

I.C. acknowledges support from the National Defense Science and Engineering Graduate Fellowship Program and the National Science Foundation (NSF) award DMR-1747426. M.S. acknowledges support from the NASA Space Technology Graduate Research Fellowship Program and the NSF Center for Integrated Quantum Materials. T.P. acknowledges support from the NSF Graduate Research Fellowship Program and the MIT Jacobs Presidential Fellowship. C.P. acknowledges support from the Hertz Foundation Elizabeth and Stephen Fantone Family Fellowship. A.J.M. acknowledges support from the Feodor Lynen Research Fellowship, the Humboldt Foundation, the MITRE Moonshot program, and the DARPA ONISQ program. This work is supported by a collaboration between the US DOE and other Agencies. This material is based upon work supported by the U.S. Department of Energy, Office of Science, National Quantum Information Science Research Centers, Quantum Systems Accelerator (QSA). This material is based upon work supported by the Air Force Office of Scientific Research under award number FA9550-20-1-0105, supervised by Dr. Gernot Pomrenke. Experiments were supported in part by the NSF Center for Ultracold Atoms (CUA). Lithium niobate fabrication at the CSEM was supported by the European Union's Horizon 2020 research and innovation program under grant agreement No. 101016138. Distribution Statement A. Approved for public release. Distribution is unlimited. This material is based upon work supported by the National Reconnaissance Office (NRO) under Air Force Contract No. FA8702-15-D-0001. Any opinions, findings, conclusions, or recommendations expressed in this material are those of the authors and do not necessarily reflect the views of the National Reconnaissance Office. ©2022 Massachusetts Institute of Technology. Delivered to the U.S. Government with Unlimited Rights, as defined in DFARS Part 252.227-7013 or 7014 (Feb 2014). Notwithstanding any copyright notice, U.S. Government rights in this work are defined by DFARS 252.227-7013 or DFARS 252.227-7014 as detailed above. Use of this work other than as specifically authorized by the U.S. Government may violate any copyrights that exist in this work. The authors thank Mikkel Heuck, Franco Wong, Artur Hermans, Alex Sludds, Saumil Bandyopadhyay, Matthew Trusheim, Eric Bersin, Kevin Chen, Marco Colangelo, Sara Mouradian, Xingyu Zhang, Nathan Gemelke, Noel Wan, Frédéric Peyskens, Dongyuu Kim, Alex Lukin, Sepehr Ebadi, Dolev Bluvstein, David Lewis, and Paul Gaudette for enlightening discussions.

## Author contributions

I.C., A.J.M., and D.E. conceived the experiments. I.C. designed systems, performed experiments, analyzed the data, and wrote the manuscript with assistance from T.P. and C.P. (software) along with M.S. and A.J.M. (experimental) and C.B. (circuit factorization). H.S., G.C., and A.G. fabricated TFLN devices. M.S. characterized the diamond sample produced by A.M., J.M., S.H., P.B.D., and D.B. D.E. and A.G. supervised the project. All authors discussed the results and contributed to the manuscript.

## Competing interests

D.E. is a scientific advisor to and holds shares in QuEra Computing. The CSEM offers lithium-niobate-on-insulator-related design, fabrication,

testing, and integration services. The remaining authors declare no competing interests.
