## [Transparent Peer Review file · Nature Communications]

An integrated photonic engine for programmable atomic control

Corresponding Author: Mr Ian Christen

Version 0:

Reviewer comments:

Reviewer #1

(Remarks to the Author)

I am very satisfied with the authors' responses to my questions and the issues raised by the other referee. I am impressed by the amendments they have made to the manuscript. Not only did they do an excellent job of highlighting the strengths of their work and placing it in context with the state-of-the-art in the field, but they also included new capabilities and discussions that appear in four new sections. Two of these sections directly address the scalability of their platform, which was one of my main concerns. They show that their methods can be extended to tens of thousands of channels. I also appreciate that they took the time to share their research directions and future plans. They are very both impressive and encouraging.

New capabilities are also presented in the figures within the main text (for example in Fig. 4), which I find very interesting and aligned with the scope of their paper.

Overall, I feel their work includes impressive technical achievements that will enable advances across different fields. The manuscript is now stronger and the results they provide are well-supported. For these reasons, I will be happy to see this work published in Nature Communications.

(Remarks on code availability)

Reviewer #2

(Remarks to the Author)

The manuscript "An integrated photonic engine for programmable atomic control" by Ian Christen et al., demonstrates a 2D array of individually controllable MZI. The system is novel and unique, and offers a path for fast optical gates across atom-like systems. The key capability that separates them from other similar technology like AOD/AOM/SLM is their GHz level bandwidth combined with arbitrary positioning/steering of the optical beams. To my knowledge these are some of the fastest switching rates demonstrated for similar systems.

The authors demonstrate this capability by combining an array of MZI with a fanout SLM for preparing the beams for the MZI and a second steering SLM that can be used for steering the beam from MZI to target position of interest within the range. This way authors are able to combine the fast switching/frequency modulation of the MZI with arbitrary positioning capability from the SLM. However, this repositioning of the beams are limited by the SLM speed. In the manuscript authors use an array of 16 MZI to generate 16 individually controlled GHz switching optical channels, with the added capability of generating more optical channels out of these 16 channels using the second SLM.

The authors demonstrate a parallel stabilization of all the MZI channels using a 4 point measurement, and homogeneity of the beams in time and space. Finally, the authors use their system capability to simultaneously readout multiple defect centers that have a random spatial distribution, and also demonstrate fast frequency tuning to readout two different defects

separated by 520 MHz. In my understanding this sort of parallel confocal pulsed excitations at nanosecond timescales is a first of its kind.

Taken individually, many of these components have been demonstrated and are well understood. And the platform is limited to ~700 nm wavelength range. However, the authors have combined them in a novel way including the introduction of the second beam shaping/steering SLM. Because of these reasons I believe this work would be of interest to the reader of Nature communication and recommend publishing.

Some comments questions that I have are:

- 1) Section V Application beginning was a bit unclear to read on whether it was a widefield excitation – individual readout vs individual excitation - widefield readout
- 2) I am confused about the some numbers. The authors note that the $f_1=f_0+520\text{MHz}$. But quoted $f_1 = 406.70750$, compared to $f_0 = 406.70906$ gives $f_1=f_0-1.5\text{GHz}$
- 3) In method 5 authors say “The opposing two cross ports of each 2x2 MZI are routed to edge couplers via waveguides threaded in between the grating waveguides “ It was not clear where and how these edge couplers are connected to the MZI, as opposed to the grating couplers.
- 4) In the main article authors say they run the MZI stabilizing algorithm at 200Hz. In the supplement authors also point out that only 2 largest deviating MZI's are able to be corrected at a single camera frame. It was not clear if in 200Hz all the MZI's are stabilized or only a part of them is stabilized.
- 5) The authors state that there is a zero point drift on the order of seconds. However the chip seems to be temperature stabilized using TEC. I expected temperature variations to be the primary cause of zero point drift. Given it is stabilized, I am curious about the origin of the drift.

(Remarks on code availability)

Reviewer #3

(Remarks to the Author)

(Remarks on code availability)

Point-by-Point Response

For this point-by-point response, *original remarks are italicized*, responses are marked in light blue, and manuscript changes are marked in blue.

Reviewer #1 (Remarks to the Author):

I am very satisfied with the authors' responses to my questions and the issues raised by the other referee. I am impressed by the amendments they have made to the manuscript. Not only did they do an excellent job of highlighting the strengths of their work and placing it in context with the state-of-the-art in the field, but they also included new capabilities and discussions that appear in four new sections. Two of these sections directly address the scalability of their platform, which was one of my main concerns. They show that their methods can be extended to tens of thousands of channels. I also appreciate that they took the time to share their research directions and future plans. They are very both impressive and encouraging.

New capabilities are also presented in the figures within the main text (for example in Fig. 4), which I find very interesting and aligned with the scope of their paper.

Overall, I feel their work includes impressive technical achievements that will enable advances across different fields. The manuscript is now stronger and the results they provide are well-supported. For these reasons, I will be happy to see this work published in Nature Communications.

We thank Reviewer #1 for their suggestions in review, which led us to put more weight towards the scaling questions that they highlighted, among other comments.

Reviewer #2 (Remarks to the Author):

The manuscript “An integrated photonic engine for programmable atomic control” by Ian Christen et al., demonstrates a 2D array of individually controllable MZI. The system is novel and unique, and offers a path for fast optical gates across atom-like systems. The key capability that separates them from other similar technology like AOD/AOM/SLM is their GHz level bandwidth combined with arbitrary positioning/steering of the optical beams. To my knowledge these are some of the fastest switching rates demonstrated for similar systems.

The authors demonstrate this capability by combining an array of MZI with a fanout SLM for preparing the beams for the MZI and a second steering SLM that can be used for steering the beam from MZI to target position of interest within the range. This way authors are able to combine the fast switching/frequency modulation of the MZI with arbitrary positioning capability from the SLM. However, this repositioning of the beams are limited by the SLM speed. In the manuscript authors use an array of 16 MZI to generate 16 individually controlled GHz switching optical channels, with the added capability of generating more optical channels out of these 16 channels using the second SLM.

The authors demonstrate a parallel stabilization of all the MZI channels using a 4 point measurement, and homogeneity of the beams in time and space. Finally, the authors use their system capability to simultaneously readout multiple defect centers that have a random spatial distribution, and also demonstrate fast frequency tuning to readout two different defects separated by 520 MHz. In my understanding this sort of parallel confocal pulsed excitations at nanosecond timescales is a first of its kind.

Taken individually, many of these components have been demonstrated and are well understood. And the platform is limited to ~700 nm wavelength range. However, the authors have combined them in a novel way including the introduction of the second beam shaping/steering SLM. Because of these reasons I believe this work would be of interest to the reader of Nature communication and recommend publishing.

We thank Reviewer #2 for their recognition of the novelty of our platform and their questions in review.

Some comments questions that I have are:

1) Section V Application beginning was a bit unclear to read on whether it was a widefield excitation – individual readout vs individual excitation - widefield readout

In the main text, we clarify that the widefield excitation is characterization done “before applying our photonic engine”. We also clarify other parts of that paragraph.

2) I am confused about the some numbers. The authors note that the $f_1=f_0+520\text{MHz}$. But quoted $f_1 = 406.70750$, compared to $f_0 = 406.70906$ gives $f_1=f_0-1.5\text{GHz}$

Thank you for finding this typo. We have corrected f_1 to 406.70854.

3) In method 5 authors say “The opposing two cross ports of each 2x2 MZI are routed to edge couplers via waveguides threaded in between the grating waveguides “ It was not clear where and how these edge couplers are connected to the MZI, as opposed to the grating couplers.

The MZIs are four port devices, two in, two out. We measure using a grating in, grating out modality in the main text. However, the other two ports of the MZI still exist, and are connected to edge couplers. Fig. S2g shows this topology. We have moved the reference to Fig. S2g to be one sentence later (now at the end of the sentence that reviewer #2 quotes) to point the reader to this clarifying image.

4) In the main article authors say they run the MZI stabilizing algorithm at 200Hz. In the supplement authors also point out that only 2 largest deviating MZI's are able to be corrected at a single camera frame. It was not clear if in 200Hz all the MZI's are stabilized or only a part of them is stabilized.

All MZIs are monitored at 200 Hz. That is, error is measured for all MZIs on each tick. However, feedback is not applied to all of the MZIs on each tick. Due to the limitations on AWG update rates, only the two parameters with the largest error are updated each tick.

5) The authors state that there is a zero point drift on the order of seconds. However the chip seems to be temperature stabilized using TEC. I expected temperature variations to be the primary cause of zero point drift. Given it is stabilized, I am curious about the origin of the drift.

We note in the main text that “Lithium niobate is known to be susceptible to zero- point drift, where the voltage corresponding to the point of highest extinction varies over time, which is attributed to fluctuations in charges trapped in the phase shifters [44–47].” Though we do not elaborate on this (it may be the subject of a separate paper) we note at the end of the same section that “We find that trajectory of zero-point drift is repeatable depending upon initial conditions, especially the average applied voltage in preceding seconds.”

Reviewer #3 (Remarks to the Author):

We thank Reviewer #3 for their contributions to our review.